# In vitro reconstitution of branching microtubule nucleation

**Ammarah Tariq, Lucy Green, J Charles G Jeynes, Christian Soeller, James G Wakefield***

Living Systems Institute, University of Exeter, Exeter, United Kingdom

**Abstract** Eukaryotic cell division requires the mitotic spindle, a microtubule (MT)-based structure which accurately aligns and segregates duplicated chromosomes. The dynamics of spindle formation are determined primarily by correctly localising the MT nucleator, γ-Tubulin Ring Complex (γ-TuRC), within the cell. A conserved MT-associated protein complex, Augmin, recruits γ-TuRC to pre-existing spindle MTs, amplifying their number, in an essential cellular phenomenon termed 'branching' MT nucleation. Here, we purify endogenous, GFP-tagged Augmin and γ-TuRC from *Drosophila* embryos to near homogeneity using a novel one-step affinity technique. We demonstrate that, in vitro, while Augmin alone does not affect Tubulin polymerisation dynamics, it stimulates γ-TuRC-dependent MT nucleation in a cell cycle-dependent manner. We also assemble and visualise the MT-Augmin-γ-TuRC-MT junction using light microscopy. Our work therefore conclusively reconstitutes branching MT nucleation. It also provides a powerful synthetic approach with which to investigate the emergence of cellular phenomena, such as mitotic spindle formation, from component parts.

*For correspondence:
j.g.wakefield@exeter.ac.uk

**Competing interests:** The authors declare that no competing interests exist.

## Introduction

Branching MT nucleation is dependent upon Augmin and γ-TuRC and generates the bulk of MTs required for both meiotic and mitotic spindle formation (*Petry et al., 2011*; *Sánchez-Huertas and Lüders, 2015*; *David et al., 2019*) and has been visualised in vivo in *Drosophila, Xenopus*, plants, and humans (*Petry et al., 2011*; *David et al., 2019*; *Hayward et al., 2014*; *Ho et al., 2011*; *Lawo et al., 2009*; *Goshima et al., 2008*; *Uehara et al., 2009*; *Prosser and Pelletier, 2017*; *Petry et al., 2013*). However, understanding, and in vitro reconstitution of, this phenomenon has been hampered by methodological constraints relating to purification of functional protein complexes (*Hsia et al., 2014*; *Song et al., 2018*); Augmin is composed of 8 subunits, while the γ-TuRC is a ~ 2 MD protein complex containing multiple copies of at least six proteins, including 14 molecules of γ-Tubulin (*Zheng et al., 1995*; *Moritz et al., 1995*; *Kollman et al., 2011*; *Tovey and Conduit, 2018*; *Oegema et al., 1999*). In vitro studies (*Oegema et al., 1999*) generally use proteins that have been individually- or co-expressed and purified in heterologous systems (*Trokter et al., 2012*; *Schlager et al., 2014*), where folding and post translational modifications crucial to function may not occur. Although purification of protein complexes from autogenous cells can be achieved using affinity-based methods, non-specific binding of contaminating proteins and difficulties in releasing purified proteins from affinity matrices are major problems.

We therefore developed an approach to allow the isolation of intact, functional Augmin and γ-TuRC, to test the hypothesis that these two complexes are necessary and sufficient for branched MT nucleation. The approach is based on biotinylated, amine-reactive thiol- or photo-cleavable linkers, Sulfo-NHS-SS-Biotin and PC-Biotin-NHS (*Figure 1a*). Stepwise incubation of the ~12 kD camelid anti-GFP nanobody, GFP-binding protein, with either of these linkers resulted in covalent linkage, while subsequent incubation with a Streptavidin Agarose matrix led to stable tri-partite reagents – GFP-

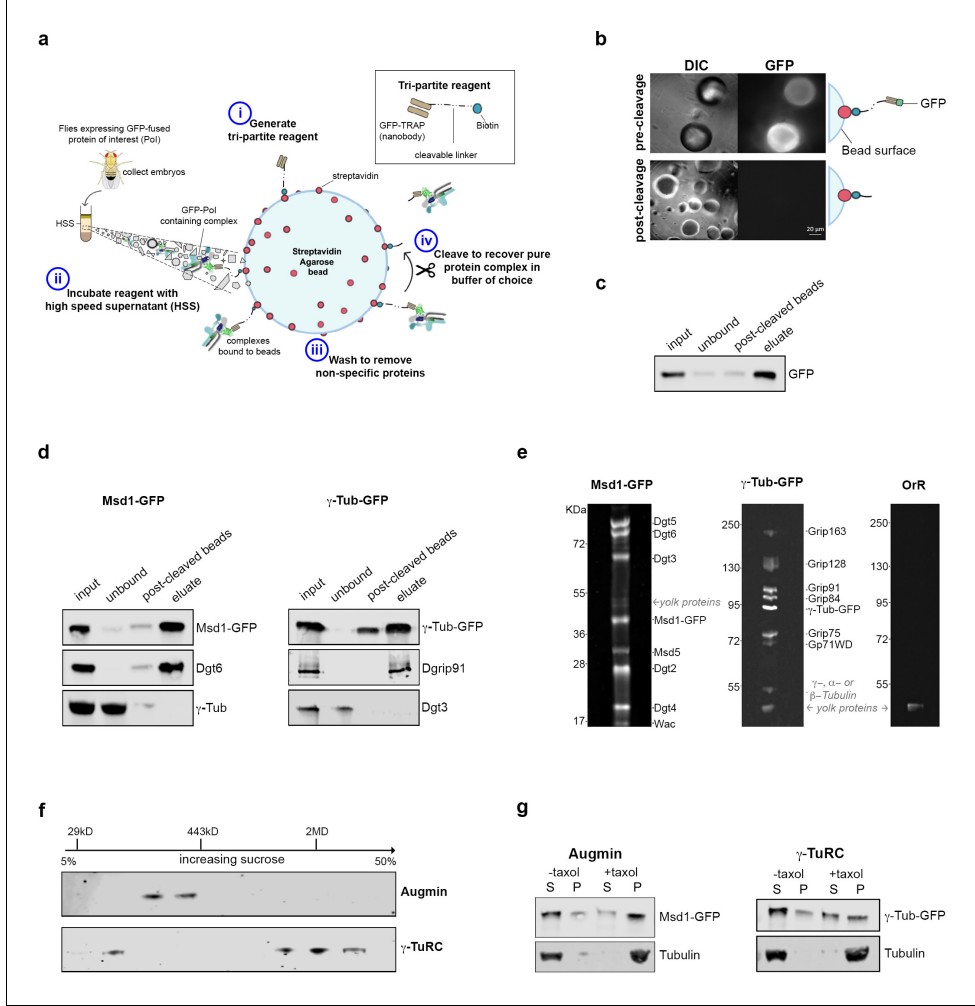

**Figure 1.** Isolation of functional γ-TuRC and Augmin using cleavable affinity purification. (**a**) Sketch of purification methodology. (**b**) Images of GFP-TRAP-Sulfo beads after incubation with His-GFP, pre- and post-cleavage by 50 mM DTT. (**c**) Western blot demonstrating the effective isolation and cleavage of soluble His-GFP. (**d**) Western blots of cl-AP of Msd1-GFP and γ-Tubulin-GFP. The fusion proteins, present in embryo extracts (input), are efficiently depleted upon incubation with GFP-TRAP-Sulfo beads (unbound), released in 50 mM DTT (post-cleaved beads) and present in the eluate. Subunits of Augmin, but not γ-TuRC, co-elute with Msd1-GFP. Subunits of γ-TuRC, but not Augmin, co-elute with γ-Tubulin-GFP. (**e**). SYPRO-ruby stained gels of post-cleaved eluates from control embryos (OrR), or MG132-treated (mitotic) embryos expressing the Augmin subunit Msd1-GFP or γ-Tubulin-GFP. (**f**) Western blot of sucrose gradient fractionation of purified mitotic Augmin or γ-TuRC. Complexes sediment as expected for their molecular weights. (**g**) Western blots of in vitro MT co-sedimentation assays. In the absence of taxol (-), Tubulin, pure Augmin (Msd1-GFP) and pure γ-TuRC (γ-Tubulin-GFP) remain in the supernatant (S). In the presence of taxol (+), Tubulin polymerises and is present in the pellet (P). Augmin and γ-TuRC co-sediment. The online version of this article includes the following figure supplement(s) for figure 1:

**Figure supplement 1.** Isolation of functional γ-TuRC and Augmin using GFP-TRAP-PC beads.

TRAP-Sulfo beads and GFP-TRAP-PC beads; where GFP-binding protein is immobilised, but cleavable through the addition of DTT or exposure to UV light, respectively (*Figure 1a*).

To test these reagents, beads were incubated with bacterially expressed and purified 6xHis-GFP, and extensively washed. Individual beads fluoresced with varying intensity and, upon brief exposure to 50 mM DTT or UV light, fluorescence decreased, concomitant with an increase in the surrounding medium (*Figure 1b*; *Figure 1—figure supplement 1a,b*). Western blot analysis confirmed >90% and~60% of GFP was released from GFP-TRAP-Sulfo and GFP-TRAP-PC beads, respectively, following cleavage (*Figure 1c*; *Figure 1—figure supplement 1c*).

We next sought to determine whether this 'cleavable Affinity Purification' (cl-AP) could be used to isolate Augmin and γ-TuRC. We have previously shown that *Drosophila* Augmin can be purified from extracts of early embryos expressing a GFP-tagged variant of the Msd1 subunit (*Chen et al., 2017*). *Drosophila* embryos have also been used to purify the γ-TuRC (*Oegema et al., 1999*; *Moritz et al., 1995*), and flies expressing γ-Tubulin-GFP are available (*Hallen et al., 2008*). As branching MT nucleation is essential during mitosis, we used embryos arrested in a metaphase-like state through incubation with the proteasomal inhibitor, MG132 (*Chesnel et al., 2006*). Both Msd1-GFP and γ-Tubulin-GFP were efficiently immobilised on GFP-TRAP-Sulfo or GFP-TRAP-PC beads and western blotting confirmed that, upon cleaving, Msd1-GFP and γ-Tubulin-GFP were concentrated in the eluate, with other subunits of the complexes co-eluting (*Figure 1d*; *Figure 1—figure supplement 1d*). To test the purity of the complexes, we subjected MG132-treated (mitotic) control (OrR), Msd1-GFP or γ-Tubulin-GFP embryo extracts to GFP-TRAP-Sulfo cl-AP followed by gel electrophoresis and SYPRO-ruby staining of eluates (*Figure 1e*). Bands corresponding to each subunit of both Augmin and γ-TuRC were identified at intensities expected for the known stoichiometric relationships between subunits (*Oegema et al., 1999*). One additional set of low intensity bands was seen in all eluates, at ~45 kD; almost certainly corresponding to yolk proteins - the most abundant proteins in *Drosophila* early embryos (*Barnett et al., 1980*). Importantly, γ-Tubulin did not co-purify with Augmin, and Dgt3 (a subunit of the Augmin complex) did not co-purify with γ-TuRC (*Figure 1d*). Moreover, sucrose gradient analysis undertaken on purified mitotic complexes determined that Msd1-GFP sedimented as expected for Augmin-GFP (~360 kD) and that γ-Tubulin-GFP sedimented in two populations – one consistent with γ-Tubulin-GFP alone and one consistent with incorporation into the γ-TuRC (2MD) (*Figure 1f*). Neither complex co-fractionated, again strongly suggesting that Augmin and γ-TuRC are purified independently of one another, or other cellular activities.

Both γ-TuRC and Augmin bind MTs in co-sedimentation assays (*Hughes et al., 2008*; *Wainman et al., 2009*; *Goshima et al., 2008*). We therefore incubated mitotic Augmin-GFP or γ-TuRC-GFP with purified Tubulin, in the presence of GTP and taxol to promote MT polymerisation, sedimenting through a glycerol cushion to separate MTs and MT associated proteins from soluble Tubulin and non-MT binding proteins (*Figure 1g*; *Figure 1—figure supplement 1e*). As expected, both Msd1-GFP and γ-Tubulin-GFP co-sedimented with MTs, demonstrating purified Augmin and γ-TuRC maintain at least some of their cellular properties.

To assess the effects of purified Augmin and γ-TuRC on MT nucleation and polymerisation, we used a highly-reproducible quantitative assay, where incorporation of a dye into MTs as they polymerise is measured as a change in fluorescence (*Bonne et al., 1985*) (Cytoskeleton Inc). Incubation of Tubulin in the presence of GTP and glycerol at 37°C resulted in its polymerisation over ~1 hr, with sigmoidal dynamics corresponding to lag, nucleation, polymerisation and plateau phases (*Figure 2a*; *Figure 2—figure supplement 1*). The time at which 50% of polymerisation was achieved (x50) was 31.5mins (± 0.5 mins) (*Figure 2b*). Addition of purified γ-TuRC-GFP stimulated MT nucleation, causing a shift in the polymerisation curve and a reduction in the x50 to 16.5 mins (± 1.2 min) (*Figure 2a, b*), confirming its functionality. In contrast, addition of purified Augmin-GFP had no significant effect on the shape of the polymerisation curve or the x50 (32.5 mins (± 1.5 min) (*Figure 2a,b*). Therefore, although Augmin-GFP binds MTs it does not, in isolation, change MT nucleation/polymerisation dynamics. However, addition of Augmin-GFP dramatically enhanced γ-TuRC-dependent nucleation of MTs, further reducing the x50 to 9.5 min (± 0.45 min) (*Figure 2a,b*). This effect was specific for the physical interaction between Augmin and γ-TuRC, as addition of bacterially expressed and purified truncated Augmin subunits, Dgt3, Dgt5 and Dgt6, which we previously demonstrated interact directly with γ-TuRC (*Chen et al., 2017*), resulted in nucleation/polymerisation curves indistinguishable to γ-TuRC alone (*Figure 2—figure supplement 1*). Thus, purified Augmin does, indeed, augment γ-TuRC-dependent MT nucleation in vitro.

Importantly, this phenomenon was dependent on whether the complexes were isolated from mitotically arrested (MG132) or untreated, mainly interphase (cycling) embryos. γ-TuRC purified from cycling embryos was a less efficient nucleator than its mitotic counterpart (*Figure 2c*; *Figure 2—figure supplement 2*), while cycling Augmin, like mitotic Augmin, showed no independent activity. However, when incubated together, either mitotic γ-TuRC and cycling Augmin or cycling γ-TuRC and mitotic Augmin were able to enhance MT nucleation to the same extent as both mitotic complexes (*Figure 2c*; *Figure 2—figure supplement 2*). As cell cycle dependent changes in protein

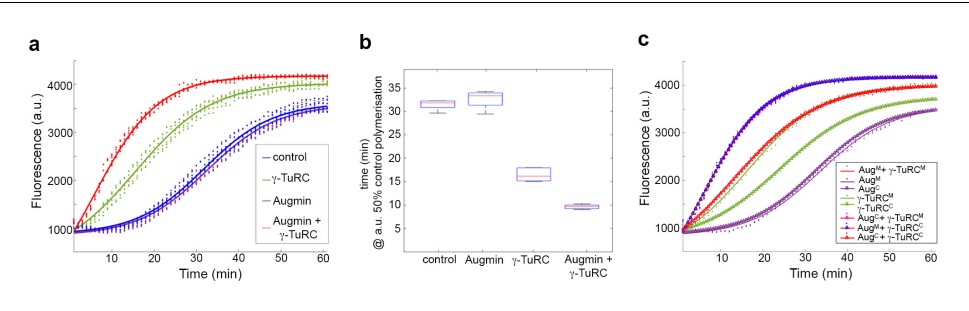

**Figure 2.** Pure Augmin enhances γ-TuRC-dependent MT nucleation in a cell cycle dependent manner. (**a**) Tubulin polymerisation assays, where fluorescence is directly related to the amount of Tubulin polymer present. The curves are a sigmoidal fit to six replicates of the experiment (dots); three independent purification experiments, each undertaken in duplicate. (**b**) Plot of the x50 in relation to control polymerisation assay, showing the median (red line), interquartile ranges (blue box) and 95% confidence intervals of the median (notches) for each condition. The differences in the time taken for 50% polymerisation between all conditions differs significantly at p=0.001, except when comparing control to Augmin (ANOVA). (**c**) Fluorescent tubulin polymerisation assays undertaken with complexes isolated from MG132-treated (mitotic) or cycling embryos.

The online version of this article includes the following figure supplement(s) for figure 2:

**Figure supplement 1.** The synergistic effect of pure Augmin on γ-TuRC-dependent MT nucleation in vitro is dependent on the Augmin-γ-TuRC interface.

**Figure supplement 2.** Individual sets of polymerisation curves using Augmin and γ-TuRC purified from either mitotic (M) or cycling (C) embryo extracts, taken from *Figure 2c*.

---

function are determined mostly by post-translational modifications (PTMs), this observation suggests that PTMs of either Augmin or γ-Tubulin are crucial in vivo.

To characterise the morphology of MTs generated in the presence of purified mitotic Augmin and γ-TuRC, we initially took samples from the in vitro assays at t = 15 min, fixing and imaging them via fluorescence microscopy. Control samples, or those containing purified Augmin, showed only very few, short MTs per field of view while, as expected, γ-TuRC-containing samples possessed many individual MTs (*Figure 3a*). In contrast, samples incubated simultaneously with γ-TuRC and Augmin showed extensive MT branching, bundling and nesting, rather than individual MTs (*Figure 3a*). Moreover, consistent with the observation that, in vivo, Augmin is required for γ-Tubulin localisation to the mitotic spindle, purified mitotic Augmin was able to recruit purified γ-TuRC to MTs in vitro, co-localising along the length of MTs in distinct punctae (*Figure 3—figure supplement 1a*). To temporally resolve the MT nucleation and polymerisation in the presence of Augmin and γ-TuRC, we fixed reactions at earlier timepoints. Even within 1 min of the simultaneous addition of Augmin and γ-TuRC, MT nucleation and putative branching were observed, though background fluorescence of soluble tubulin was high (*Figure 3b*). Samples taken at t = 2 and t = 5 min showed similar numbers of MTs to each other, per field of view, though the mean length of MTs, and proportion of MTs with branches, were significantly greater after 5 min, than after 2 min (4.01 μm versus 3.24 μm and 34% versus 24%, respectively) (*Figure 3b*; *Figure 3—source data 1*). This suggests that MT nucleation from purified γ-TuRCs reaches maximal activity within the first few minutes of the assay. Moreover, the estimated number of γ-TuRCs per coverslip ($6.0 \times 10^6$) was 2-3 times less than the mean number of MTs per coverslip ($1.7 \times 10^7$) (see Materials and methods). These calculations suggest either (i) that our approximations of γ-TuRC molarity are out by a factor of 2-3, (ii) that the MTs counted include a proportion that have been nucleated by a γ-TuRC and subsequently broken or (iii) that a single γ-TuRC is able to nucleate multiple MTs at a rate of >1 MT/min, perhaps by nucleation and subsequent release.

Finally, we sought to conclusively reconstitute and visualise the branching MT-Augmin-γ-TuRC-MT junction, and to measure the ability of the purified γ-TuRCs to simultaneously bind MT minus ends and Augmin. To do this, we generated populations of taxol-stabilised fluorescent MTs in the presence or absence of individual mitotic protein complexes, subsequently co-incubating them. Co-incubation of HiLyte Fluor 488-MTs ([488]MTs) with Rhodamine-MTs ([Rhod]MTs) resulted in independent populations of MTs. Occasional examples of [Rhod]MTs terminating at [488]MTs were observed, but at a

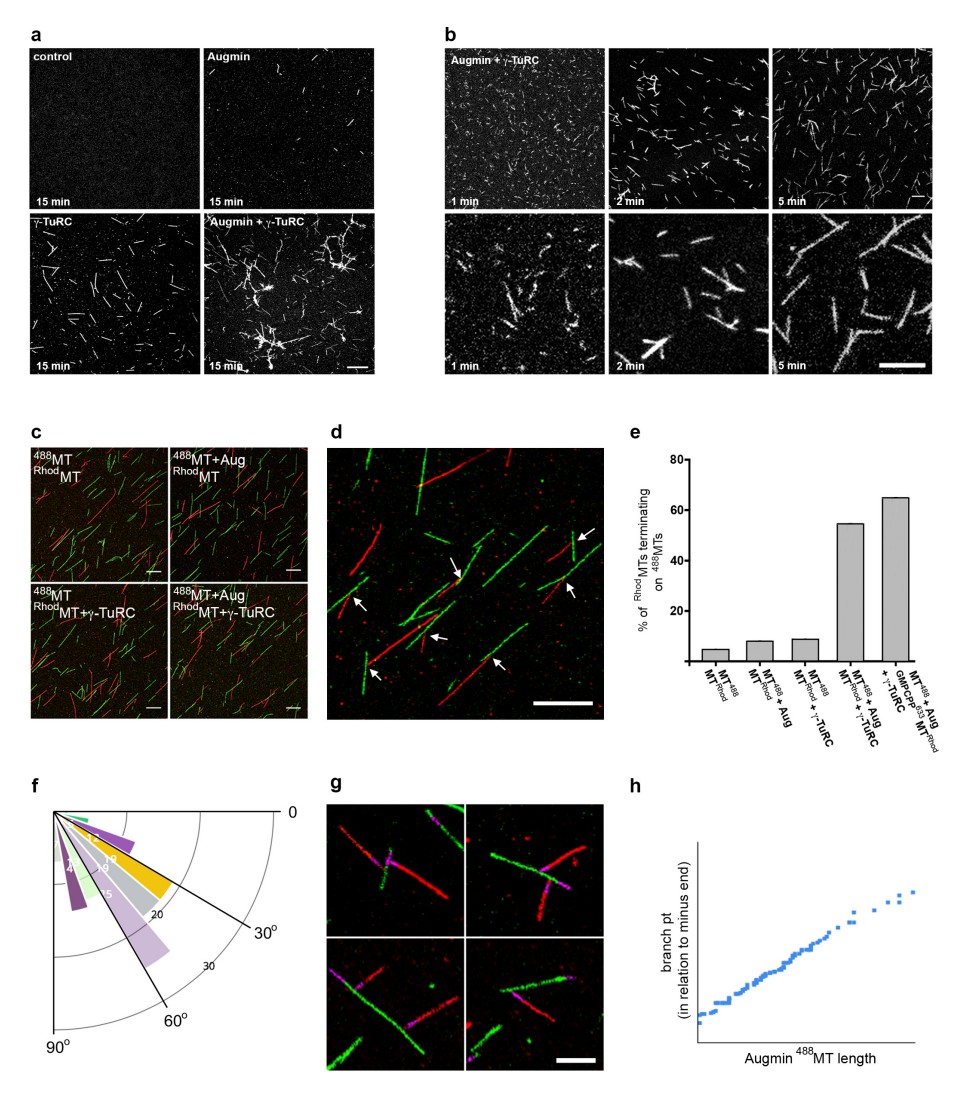

**Figure 3.** Reconstitution of the MT-Augmin-γ-TuRC-MT junction. (**a**) Confocal images of fixed, fluorescent polymerisation assays at t = 15 mins for each condition. (**b**) Confocal images of fixed, fluorescent polymerisation assays at t = 1, 2 and 5 mins in the presence of purified Augmin and γ-TuRC. (**c**) Confocal images of taxol stabilised MTs, formed in the absence or presence of purified Augmin and γ-TuRC, and co-incubated. Incubation of Augmin-488MTs (green) with γ-TuRC-RhodMTs (red) leads to MT branches. (**d**) Higher magnification of the MT-Augmin-γ-TuRC-MT junctions (arrows). (**e**) Histogram of the percentage of RhodMTs that terminate precisely at a 488MT, generating a branchpoint. (**f**) Distribution of γ-TuRC-RhodMTs junction angles, in relation to Augmin-488MTs. (**g**) Confocal images of junctions, using GMPCPP seeds (purple) to distinguish the MT minus ends. Minus ends of γ-TuRC-RhodMTs (red) interact with Augmin-488MTs (green). (**h**) Pearson correlation coefficient plot demonstrating a strong positive correlation between the length of Augmin-488MTs and the position of γ-TuRC-RhodMT branch points. Scale bars, a-d, 5 μm; scale bar, g, 2 μm.

The online version of this article includes the following source data and figure supplement(s) for figure 3:

**Source data 1.** Excel spreadsheets of the individual data points corresponding to *Figure 3b* (number and length of MTs per field of view at t = 2 and t = 5 mins); *Figure 3d* (Histogram of % of Rhod-MTs terminating on a 488-MT – i.e. % of γ-TuRC activity); *Figure 3e* (diagram of branch angles); *Figure 3f* (branchpoint distance from MT minus end, as assessed by GMPCPP seed).

**Figure supplement 1.** Augmin recruits γ-TuRC to MTs.

frequency consistent with random placement of MTs on coverslips (~6%) (*Figure 3b,d*; *Figure 3— source data 1*). Similar results were obtained with fluorescent MTs incubated with either Augmin or γ-TuRC (*Figure 3c,e*; *Figure 3—source data 1*). However, when [Rhod]MTs incubated with γ-TuRC were added together with [488]MTs incubated with Augmin, 55% of γ-TuRC-containing [Rhod]MTs terminated precisely at an Augmin-containing [488]MT (*Figure 3c–e*; *Figure 3—source data 1*). Although the polarity and angles of the [Rhod]MTs branches in relation to the 'mother' [488]MTs varied far more widely than seen in vivo, presumably due to how the MTs settled on the coverslip during preparation, we found a bias in the angle, similar to that seen in living *Drosophila* (*Verma and Maresca, 2019*) (*Figure 3e*; *Figure 3—source data 1*). To unequivocally demonstrate the polarity of the MTs, and to take into account MTs that might have lost their native γ-TuRC-bound minus end during sample preparation, we formed these junctions in the presence of pre-stabilised HiLyte 647-labelled GMPCPP MT seeds. As expected, the polarity of the interaction between γ-TuRC-[Rhod]MTs and Augmin-[488]MTs was specific; with only the minus ends of γ-TuRC-[Rhod]MTs precisely terminating at the Augmin-[488]MTs lateral surfaces (*Figure 3g*). Moreover, as predicted, the percentage of γ-TuRC-[Rhod]MTs generating branches further increased to 64% (*Figure 3e*; *Figure 3—source data 1*). Finally, we found that the formation of junctions on Augmin-containing [488]MTs was length independent – branches were equally distributed at the very distal MT tips, throughout their length and on the GMPCPP seed itself (*Figure 3g,h*; *Figure 3—source data 1*).

Our work confirms a long-standing hypothesis, first articulated to explain the loss of γ-Tubulin on the mitotic spindle when the expression of Augmin subunits is reduced (*Goshima et al., 2007*). It demonstrates conclusively that Augmin directly recruits γ-TuRC to MTs and that these two protein complexes are sufficient for robust and efficient branching MT nucleation. Although our calculations suggest that 65% of the mitotic γ-TuRCs are competent to bind MT minus ends and Augmin, and that they are able to nucleate at a rate of >1 MT/min, we expect that, in vivo, other proteins further enhance these γ-TuRC activities. For example, a clear role has been reported for the MT associated protein, TPX2, in stimulating Augmin-dependent branched MT nucleation in *Xenopus* meiotic extracts (*Petry et al., 2013*; *Thawani et al., 2019*). However, in support of our conclusions, *Drosophila* TPX2 has recently been shown to be dispensable for the phenomenon in vivo (*Verma and Maresca, 2019*). Our experiments also highlight the intriguing possibility that, in some cells, Augmin might recruit pre-existing γ-TuRC-containing MTs, nucleated elsewhere in the cell, anchoring them to specific sites and increasing local MT density.

The generation of stable MT-Augmin-γ-TuRC-MT junctions using the methodologies pioneered here also provide a route to finally defining the molecular detail of MT branching at the ultrastructural level. More broadly, cleavable affinity purification provides the basis to generate more complex, but molecularly defined, mixes of purified proteins, complete with in vivo PTMs, in order to reconstitute higher-order aspects of spindle formation. Indeed, by isolating and combining purified, active proteins and protein complexes from any biological system of interest, "cl-AP TRAP" overcomes the limitations of traditional 'bottom-up' approaches, allowing exploration of the level of biological organisation between individual protein and biological process – the level at which emergence of cellular phenomena often occurs.

# Materials and methods

## *Drosophila* husbandry and embryo collection

Flies were kept on standard medium and grown according to standard laboratory procedures. w1118 or Oregon R (OrR) strains (Bloomington Stock Center) were utilized as a wild-type controls. Transgenic flies used were: pUAS-Msd1-GFP; maternal-α-Tubulin-Gal4 (*Bonne et al., 1985*) and pNcd(γ−Tub37C-GFP), a gift from Sharyn Endow (*Chesnel et al., 2006*). 0–3 hr old embryos were collected from apple juice/Agar plates, dechorionated using bleach, washed, flash frozen in liquid nitrogen and stored at −80℃. For MG132 treatment, embryos were incubated in a solution containing 66.5% PBS (Melford), 33.2% heptane (Sigma) and 0.3% MG132 (Sigma) for 20 mins, prior to rinsing and flash freezing as above. Each embryo collection resulted in 0.05–0.1 g of frozen embryos, different batches of which were combined for each biochemical assay/purification. Routinely, batches of MG132-treated embryos were fixed and stained to visualise chromosomes and MTs, to verify their mitotic arrest.

## Preparation of GFP-TRAP-PC and GFP-TRAP-Sulfo beads

To prepare GFP-TRAP-Sulfo beads, 200 µL of GFP-binding protein (1 mg/mL) (Chromotek) in PBS was incubated with 86.1 µl of Sulfo-NHS-SS-Biotin (sulfosuccinimidyl-20(biotinamido)ethyl-1,3-dithio-propionate) (8 mM) (Thermo Scientific) for 2 hr at 4°C. Unreacted linker was removed by desalting. 40 µL of High Capacity Streptavidin Agarose Resin (Pierce) was washed 3 times for 5 mins in PBS and incubated with the GFP-binding protein-SS-Biotin product for 1 hr at 4°C with gentle rotation. The beads were washed 3 times for 5 mins in BRB80 + 0.1% IGEPAL, the volume of BRB80 was reduced to 40 µL (to maintain a 50% bead slurry) and beads stored at 4°C for use. A similar protocol was used to generate GFP-TRAP-PC-Biotin-NHS beads, but using 50 µL of GFP-binding protein, incubated with 1.45 µL of PC-Biotin-NHS (50 mM) (Ambergen) for 40 min at RT with gentle rotation and 400 µL of standard Streptavidin-Agarose bead slurry (Sigma Aldrich). Over the course of the project, different conditions were tried in an attempt to increase the photo-cleaving of GFP, Augmin-GFP and γ-TuRC-GFP from the GFP-TRAP-PC beads. We were unable to consistently reach cleaving efficiencies of greater than 60%. In contrast, cleavage of proteins and complexes from GFP-TRAP-Sulfo-beads reproducibly gave efficiencies of >90% (n => 10).

## Cleavable affinity purification

*Drosophila* embryo High Speed Supernatant (HSS) was prepared using batches of frozen 0–3 hr embryos. Embryos were dounce homogenised in BRB80 + 0.1% IGEPAL + 1 mM PMSF (Sigma), PhosSTOP phosphotase inhibitors and cOmplete, Mini, EDTA free protease inhibitors (Roche) at a ratio of 100 mg embryos to 200 µL buffer. Extracts were clarified through centrifugation at: 17,000 g for 10 min, 100,000 g for 10 min, and 100,000 g for a further 30 min. 6xHis-GFP was purified from bacteria expressing pQE80-His-GFP, using standard protocols and HisPur Cobalt Resin (Thermo Scientific), and dialysed into BRB80 + 0.1% IGEPAL + 1 mM PMSF.

Typically, HSS from 200 mg embryos or 6xHis-GFP (100 ng) were incubated overnight at 4°C with 20 µL of either GFP-TRAP-PC or GFP-TRAP-Sulfo beads. For SYPRO-Ruby staining (see below) HSS from 1 g of embryos expressing either Msd1-GFP or γ-Tubulin-GFP were used, incubated with 50 µL of GFP-TRAP-Sulfo beads. Beads were centrifuged at 2500 g for 2 mins and the depleted supernatant removed and discarded, or used for Western blotting. For cleaving of Sulfo-beads, beads were washed 4 times for 5 mins in BRB80 + 0.1% IGEPAL, resuspended in 25 µL of BRB80 + 0.1% IGEPAL, prior to addition of 25 µL of 100 mM DTT in BRB80 0.1% IGEPAL for 5 mins (50 mM final concentration). For PC-beads, beads were washed as above, then transferred to a microscope slide with a concave cavity with 25 µL of BRB80 0.1% IGEPAL. The slide was placed 10 cm below a UV lamp (UVP XX-15L (Analytik Jena US)) and exposed for 30 s intervals with gentle mixing between. Eluates were removed using a Gel-Saver II pipette tip (STARLAB). Cleaved beads were washed 4 times for 5 min in BRB80 0.1% IGEPAL. After each purification, samples were taken for SDS-PAGE and Western blotting alongside known amounts of 6xHis-GFP, in order to quantify yield and concentration of purified complexes.

## Calculation of the molarity of augmin and γ-TuRC following cl-AP

### Estimation of the concentration of GFP

Western blotting from 6 independent experiments estimates that cl-AP from ~200 µg of either Msd1-GFP or γ-Tubulin-GFP 0–3 hr embryo extract results in soluble complexes containing the equivalent of 8–9 ng of His-GFP per 1 µl. His-GFP has a molecular weight of 28 kD, therefore one mole of GFP weighs 28000 g. Given number of molecules in a mole is $6.02 \times 10^{23}$, 8.5 ng of GFP contains $1.8 \times 10^{11}$ molecules of GFP. The number of moles of GFP in 1 litre is therefore ($1.8 \times 10^{11} \times 1 \times 10^{6}$) and the molar concentration is 300 nM.

### Estimation of the concentration of purified Augmin

Sucrose gradient sedimentation followed by western blotting demonstrates that all Msd1-GFP in the purified supernatant isolated from GFP-Msd1 expressing embryos is present in fractions consistent with incorporation into Augmin (*Figure 1f*). Therefore, the number of Augmin complexes in 1 µl of cl-AP supernatant, isolated from GFP-Msd1 expressing embryos, is $1.8 \times 10^{11}$ and the molar concentration of Augmin is 300 nM. The final concentration of Augmin in the MT co-sedimentation assay

was 60 nM (10 µl in 50 µl total), while the final concentration of Augmin in the MT polymerisation assays was 30 nM (5 µl in 50 µl total).

## Estimation of the concentration of purified γ-TuRC

cl-AP from GFP-γ-Tubulin expressing embryos resulted in a mixed population of GFP-γ-Tubulin that is either incorporated into a γ-TuRC, or monomeric (partially folded, or folded but not incorporated). Sucrose gradient sedimentation followed by western blotting estimates that 60% of the GFP-γ-Tubulin in the purified supernatant isolated from GFP-γ-Tubulin expressing embryos is present in fractions consistent with incorporation into γ-TuRCs, while 40% is present in fractions consistent with γ-Tubulin that is not part of the γ-TuRC (*Figure 1f*). Purified γ-TuRCs contain ~14 molecules of γ-Tubulin[17]. As GFP-γ-Tubulin expressing embryos also contain endogenous γ-Tubulin, the γ-TuRCs isolated by cl-AP will be a mixed population. Western blotting of purified supernatant isolated from GFP-γ-Tubulin expressing embryos using anti−γ-Tubulin antibodies estimates that 75% of the γ-Tubulin is present as GFP-γ-Tubulin while 25% is present as γ-Tubulin with no tag (*Figure 1—figure supplement 1f*). Therefore, we estimate that the purified GFP-γ-TuRCs contain 8–10 molecules of GFP-γ-Tubulin (average 9) and 4–5 molecules of unlabelled γ-Tubulin. The molar concentration of γ-TuRC purified by cl-AP is therefore 300 nM x 0.6/9 = 20 nM, or the equivalent of $1.8 \times 10^{11} \times 0.6/9 = 1.2 \times 10^{10}$ γ-TuRCs per 1 µl. The final concentration of γ-TuRC in the MT co-sedimentation assay was 4 nM (10 µl in 50 µl total), while the final concentration of γ-TuRC in the MT polymerisation assays was 2 nM (5 µl in 50 µl total).

## Estimation of the nucleation efficiency of purified γ-TuRC in vitro

1 µl of polymerisation assay containing 2 nM γ-TuRC and 30 nM Augmin at t = 1, 2 or 5 min post-initiation was fixed in 200 µl glutaraldehyde fixation fix. 1 µl of this was spotted onto a 22x22 mm coverslip and imaged. We therefore estimate the total number of γ-TuRCs per coverslip as $6.0 \times 10^6$. After two mins post-initiation, an average of 130 MTs were observed per field of view (60.5 µm$^2$) (*Figure 3—source data 1*). Given there are ~$1.32 \times 10^5$ fields of view per coverslip, this equates to $1.7 \times 10^7$ MTs per coverslip. Therefore, one γ-TuRC is capable of nucleating 2–3 MTs within 2 mins.

## SDS-PAGE and western blot analysis

Protein samples were fractionated by sodium dodecyl sulfate (SDS)-polyacrylamide gel electrophoresis (PAGE). SYPRO Ruby Staining (Invitrogen) of Augmin and γ-TuRC was undertaken according to manufacturer's instructions. For Western analysis, the proteins were blotted onto a nitrocellulose membrane and probed with anti-GFP (1:1000; Sigma-Aldrich and Roche), anti-dgt6 (1:500) (a gift from M. Gatti), or anti-Dgrip91 antibodies (1:1000) (a gift from Y Zheng). IRDye 800CW goat anti-rabbit (LI-COR) and IRDye 680RD goat anti-mouse (LI-COR) IgG polyclonal Abs were used as secondary detection antibodies. Fluorescence from blots was developed with the Odyssey CLx Imaging System (LI-COR) according to the manufacturer's instructions.

## Sucrose gradient fractionation

2 ml sucrose gradients were formed by layering steps (200 µL each) of 5% to 50% sucrose in BRB80 (5% steps). 15 µL of 300 nM purified Augmin-GFP or 20 nM γ-TuRC-GFP was loaded on to each gradient and spun at 55,000 rpm for 3 hr at 4°C in a TLS-55 rotor (Beckman Coulter). Ten fractions of 180 µL were collected in each gradient from top (fraction 1) to bottom (fraction 10) and, after precipitation with 10% TCA, analysed by SDS-PAGE followed by immunoblotting for GFP. As standards, carbonic anhydrase (29 KDa), Apoferritin (443 Kda), and Blue dextran (2 MD) (Sigma) were subjected to the gradients and samples analysed by SDS-PAGE and Coomassie brilliant blue staining.

## In vitro microtubule co-sedimentation assay

Purified proteins/cl AP eluates were pre-spun at 100,000 g for 15 min at 4°C. 10 µL 300 nM GFP, 300 nM Augmin-GFP or 20 nM γ-TuRC-GFP were incubated for 15 min at 37°C in General Tubulin Buffer with 2.25 mg/mL of 99% pure porcine tubulin and 1 mM GTP (all from Cytoskeleton Inc), in a final volume of 50 µL. Taxol (Sigma) was added to 100 µM and samples incubated for a further 10 min at 37°C. A negative control for each sample was run in parallel, where samples were incubated at 4°C and where GTB replaced taxol. Samples were immediately spun through a 150 µL cushion of

BRB80 40% glycerol at 100,000 g for 45 min at 4°C in a TLA120.1 rotor (Beckmann Coulter). The supernatant and pellet fractions were analysed by Western blotting, probing with mouse anti-GFP (Roche) and anti-α-Tubulin antibodies [DM1A] (Sigma). Co-sedimentation assays were undertaken in triplicate, each with independently purified Augmin or γ-TuRC. A representative western is shown.

## MT polymerization assays

MT polymerization assays were performed using a fluorescence-based Tubulin Polymerization kit (Cytoskeleton Inc, Denver CO, Cat. # BK011P) following the manufacturer's instructions. Briefly, 5 µL of 300 nM GFP, 300 nM Augmin-GFP or 20 nM γ-TuRC-GFP was pipetted into wells within a 96-well microtiter plate, followed by 45 µL of Tubulin Reaction Mix. Tubulin polymerization was initiated by transferring the plate to a 37°C chamber of a plate reader. The polymerization dynamics of Tubulin were monitored for 60 min at 37°C by measuring the change in fluorescence every 1 min using a TECAN infinite 200pro fluorimeter, at excitation of 350 nm and emission of 440 nm. Assays to assess the effect of purified Augmin and γ-TuRC presented in *Figure 2A* are the summation of 3 individual purifications of each protein complex, undertaken in duplicate wells (six data points). Assays to assess the difference in polymerization between cycling and MG132 purified Augmin and γ-TuRC, presented in *Figure 2C* are the summation of triplicate experiments. In vitro polymerization assays to assess the difference in Augmin-dependent polymerization upon addition of competing truncated Augmin subunits (*Figure 2—figure supplement 2*) are the summation of triplicate experiments.

## Generation of fluorescent MTs

Tubulin Polymerization assays were performed as described above, but with the following modifications: Rhodamine- or HiLyte Fluor 488-tubulin was used (Cytoskeleton, Inc, Denver, CO) at a 1:10 ratio with unlabelled porcine tubulin (final tubulin concentration, 2 mg/ml). At t = 1, 2, 5 or 15 min, 1 µl of polymerisation sample was added to 200 µl glutaraldehyde (0.5% final concentration in GTB + 1 mM GTP) and incubated at RT for 15 mins. 1 µl of these fixed MTs *were spotted onto glass coverslips* and imaged.

To generate fluorescent, taxol-stabilised MTs, Rhodamine- or HiLyte Fluor 488-tubulin was polymerized as above, but in the absence of purified complexes, but in the presence of taxol at a final concentration of 20 µM. After 7 mins, the taxol-stabilized MTs were removed from the well and incubated with 5 µL of 300 nM Augmin-GFP, 20 nM γ-TuRC-GFP or buffer for 10 min at room temperature. Samples were then layered over a 150 µL cushion of 15% glycerol in BRB80 and centrifuged at 135,000 × g for 10 mins at 25°C in a TLA120.1 rotor (Beckmann Coulter). The MT pellets were gently resuspended in 50 µL of General Tubulin Buffer containing 1 mM GTP and 30% glycerol, combined as necessary as 1:1 mixtures and incubated for 10 min at room temperature. The samples were fixed with glutaraldehyde (0.5–2% final concentration) and spotted onto glass coverslips as above. Images were taken from randomly distributed fields of the coverslips using a Leica TCS SP8 confocal laser scanning microscope. Representative images from one of three independent experiments are shown.

For experiments to visualise MT polarity, HiLyte 647-labelled GMPCPP MTs were generated according to the manufacturer's instructions (Jena Biosciences). Seeds were added to the polymerisation reactions at a 1:10 ratio and processed as described above. Images were taken from randomly distributed fields of the coverslips using a Leica TCS SP8 confocal laser scanning microscope. Representative images from one of two independent experiments are shown.

## Imaging and image analysis

Samples were imaged using an inverted Leica TCS SP8 confocal laser scanning microscope using a HCOL APO CS2 63x, NA 1.4 oil immersion lens (Leica, Wetzlar, Germany). Standard filter sets were used to visualize Rhodamine, HiLyte Fluor 488, and HiLyte Fluor 647 fluorescence. Images were captured as TIFFs using Leica Application Suite X (LAS X), opened in FIJI and levels adjusted to maximise the full range of pixel intensities. To estimate the number of MTs nucleated by purified γ-TuRC and Augmin at different timepoints during the polymerization assay, images were despeckled, a duplicate, segmented mask (otsu method) applied and a minimum size threshold of 0.4 µm (to filter non-MT background) was set. After skeletonizing, the average length of the identified MTs was calculated for 10 fields of view at both t = 2 and t = 5. To quantify the % of $^{488}$MTs with $^{Rhod}$MT

branches, 8 fields of view of each sample were randomly captured. $^{488}$MTs over ~1 µM length in each field of view were totaled, alongside the number of $^{Rhod}$MTs whose fluorescence terminated precisely at a HiLyte$^{488}$MT lateral surface (between 243–339 $^{488}$MTs per sample condition). To quantify the position of the branchpoints relative to the minus ends of MTs, the length of HiLyte$^{647}$ GMPCPP-containing $^{488}$MTs were measured using the line tool in ImageJ. The position of the $^{Rhod}$MT branch from the minus end of these MTs, was also measured. The two sets of variables were subjected to Pearson correlation coefficient analysis using an on-line tool (https://www.socscistatistics.com/tests/pearson/default2.aspx), providing the graph in *Figure 3—figure supplement 1*. The Pearson correlation coefficient was r = 0.9962 at a p value of < 0.00001.

## Statistical analysis of polymerisation curves

Polymerisation data sets were fitted using a sigmoidal function. This characterised each data set in terms of the maximum and minimum fluorescence, the x value (min) at 50% distance between maximum and minimum y fluorescence value (termed 'x50') and the slope. The mean x50 value for the control samples (n = 6) was x = 31.99 (minutes) and y = 2411 (fluorescence a.u.). This value was used as a fix point to compare all the data sets. A one-way ANOVA was performed to compare the four variables (control, Augmin, γ-TuRC, and Augmin+γ-TuRC), with each variable having six repeats. A *post-hoc* Tukey test showed that all variables were significantly different from each other at p=0.001 apart from the control and Augmin, which were not significantly different from each other. Similar analyses were undertaken for the comparison of polymerisation datasets between complexes isolated from MG132 and cycling embryos, and the augmenting activity of Augmin in the presence of truncated Dgt3, Dgt5 and Dgt6 proteins. All fitting and analyses were performed in MATLAB. The code and data can be found at http://www.github.com/charliejeynes/microtubules (*Jeynes, 2020*; copy archived at https://github.com/elifesciences-publications/microtubules).

## Acknowledgements

We thank Stephen Green and Mark Wood (Exeter, UK) for initial discussions about photocleavable and other tags, and Mary Munson (Massachusetts, US) for highlighting the potential of thiol-cleavable tags for native complex purification, as published by the Rout lab (Rockefeller, US) (*Fridy et al., 2015*). We thank Chromotek for the gift of GFP-binding protein; Jack Chen, Dan Hayward and Chris Sullivan for initial attempts to develop the cl-AP technique; Marwan Al-Maqtoofi who supervised a set of University of Exeter Natural Sciences undergraduate students and the Carlota Palmer PhD students in setting up the plate reader-based MT nucleation/polymerisation assays. We thank Carolyn Moores (Birkbeck, UK), Iris Leuke and Thomas Surrey (Crick Institute, UK) for discussions and advice, and Sabine Petry for sharing unpublished data, critical reading of the manuscript and for co-ordinating submissions of manuscripts to eLife. Finally, we thank the reviewers, who made very helpful and timely suggestions to improve the final article. LG was supported by a University of Exeter Proof of Concept Award, AT by a Carlota Palmer University of Exeter PhD Studentship and JCGJ by an Innovation Fellowship supported by the Science and Technology Facilities Council (STFC), UK.

## Additional information

### Funding

| Funder | Grant reference number | Author |
|---|---|---|
| University of Exeter | Proof of Concept | Lucy Green |
| University of Exeter | Carlota Palmer PhD studentship | Ammarah Tariq |
| Science and Technology Facilities Council | Innovation Fellowship | J Charlie Jeynes |

The funders had no role in study design, data collection and interpretation, or the decision to submit the work for publication.

## Author contributions

Ammarah Tariq, Formal analysis, Validation, Investigation, Visualization, Methodology, Writing—original draft, Writing—review and editing; Lucy Green, Formal analysis, Validation, Investigation, Methodology; J Charles G Jeynes, Software, Formal analysis; Christian Soeller, Supervision, Writing—review and editing; James G Wakefield, Conceptualization, Supervision, Funding acquisition, Methodology, Writing—original draft, Project administration, Writing—review and editing

## Author ORCIDs

Christian Soeller [iD] http://orcid.org/0000-0002-9302-2203
James G Wakefield [iD] https://orcid.org/0000-0003-3616-2346

## Decision letter and Author response

Decision letter https://doi.org/10.7554/eLife.49769.sa1
Author response https://doi.org/10.7554/eLife.49769.sa2

# Additional files

## Supplementary files

• Transparent reporting form

## Data availability

All data generated or analysed during this study are included in the manuscript and supporting files.

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
