## [Decision Letter]

**Acceptance summary:**

Branching microtubule nucleation from the lateral surface of pre-existing microtubules has emerged as an important nucleation mechanism contributing, for example, to mitotic and meiotic spindle assembly and to microtubule organization in post-mitotic neurons. In this manuscript the authors show that the previously identified main players, augmin and γ-TuRC, purified as native protein complexes from *Drosophila* embryo extract, are sufficient to reconstitute branching nucleation in vitro. It is further shown that augmin stimulates γ-TuRC nucleation activity in a cell cycle-dependent manner. These are important conceptual advances in our understanding of fundamental cell biological processes. By establishing the minimally required components and revealing a regulatory activity, this manuscript also provides the framework and tools for further analyses in future studies.

**Decision letter after peer review:**

Thank you for submitting your article "in vitro reconstitution of branched microtubule nucleation" for consideration by *eLife*. Your article has been reviewed by three peer reviewers, including Jens Lüders as the Reviewing Editor and Reviewer #1, and the evaluation has been overseen by Anna Akhmanova as the Senior Editor.

The reviewers have discussed the reviews with one another and the Reviewing Editor has drafted this decision to help you prepare a revised submission.

Summary:

The paper "in vitro reconstitution of branched microtubule nucleation" by Tariq et al. presents the biochemical reconstitution of microtubule nucleation from the lattice of pre-existing microtubules. Using affinity purification from *Drosophila* embryo extract the study suggests that purified augmin and γ-TuRC, two multi-subunit protein complexes previously implicated in branching nucleation, are sufficient for this activity.

The reconstitution of branching nucleation from purified proteins is a major achievement. A weakness of the study is that the visualization of branched nucleation in fixed samples is not very convincing and that the authors have never looked at the branching nucleation process by real-time imaging; it would have been a doable experiment assuming the authors had access to a TIRF microscope. On the other hand, all reviewers agree that a unique aspect of this study is that, by collecting embryos in mitosis and interphase, the authors directly compared the augmin and γ-TuRC activities of both phases. This is not easily achievable by approaches utilizing recombinant proteins. Moreover, the finding that augmin stimulates γ-TuRC dependent nucleation would also be an important contribution to our understanding of branching nucleation.

Essential revisions:

In summary, while the study represents an important advance, it is not fully convincing in its current form. The authors need to improve the quality of the branching nucleation assay, include quality controls for the purified protein complexes and provide an estimate of the efficiency of the reconstitution. This is crucial since reconstitution with low efficiency may indicate that an important factor is missing.

1) The demonstration of reconstituted branching nucleation in Figure 3A needs to be improved. Tubulin appears to form aggregates rather than clear branches. If real-time imaging cannot be provided, another possibility is to perform the assay on pre-assembled MTs. These could be labeled in one color and then, after addition of augmin and γ-TuRC, the nucleated branches could be labeled with tubulin of a different color.

2) Throughout the manuscript, the authors refer to "pure" proteins, for example "pure Augmin." However, none of the protein reagents are pure as would be revealed by analysis by mass spectrometry. Therefore "pure" should be replaced with "purified" or "enriched." Following the same reasoning, the statement – "It demonstrates conclusively that Augmin directly recruits γ-TuRC to MTs and that these two protein complexes are sufficient for the phenomenon of branched MT nucleation." cannot be made. This issue could be potentially resolved by demonstrating a high efficiency of the reconstitution (see point 3 below) and by ensuring the quality of the purified complexes. The authors should at least show that they purify complexes of the expectd size (e.g. by gel filtration or sucrose gradient analysis) and also probe purified augmin with γ-TuRC antibodies and purified γ-TuRC with augmin antibodies to exclude significant co-purification. It is also important to compare size and composition of preparations from the two different cell cycle stages that are used in the assays. Gels are only shown for "interphase" augmin and γ-TuRC.

3) Please indicate protein concentrations used in each experiment in a reader-friendly manner (provide nM or µM instead of providing volume and mass). The authors should provide an estimate of the efficiency of the branched nucleation, the fraction of γ-TuRC that is active.

[Editors' note: further revisions were requested prior to acceptance, as described below.]

Thank you for resubmitting your work entitled "in vitro reconstitution of branching microtubule nucleation" for further consideration at *eLife*.

The reviewers have evaluated your revisions and have noted that an important issue raised in the first round of review was not addressed. It concerns the estimation of the specific activity of gTuRC in the branching nucleation assay (assay in Figure 3A). As outlined in the previous decision letter and in the follow-up clarification, the reviewers feel that without this information it is difficult to assess if the reconstituted system is complete. It seems that in your revision you have provided an alternative quantification, the efficiency of branch site formation from preformed MT/augmin and MT/gTuRC complexes. However, this is not a measure of the gTuRC nucleation activity.

Before coming to a decision, we would like to ask if you can obtain this data and add it to your revised manuscript?

---

## [Author Response]

Essential revisions:In summary, while the study represents an important advance, it is not fully convincing in its current form. The authors need to improve the quality of the branching nucleation assay, include quality controls for the purified protein complexes and provide an estimate of the efficiency of the reconstitution. This is crucial since reconstitution with low efficiency may indicate that an important factor is missing.1) The demonstration of reconstituted branching nucleation in Figure 3A needs to be improved. Tubulin appears to form aggregates rather than clear branches. If real-time imaging cannot be provided, another possibility is to perform the assay on pre-assembled MTs. These could be labeled in one color and then, after addition of augmin and γ-TuRC, the nucleated branches could be labeled with tubulin of a different color.

We were unable to obtain live image series of sufficient quality using our current TIRF set up and methodology (probably due to technical considerations, such as required surface passivation and optimisation of fluorescent tubulin concentrations). We therefore spent significant time over the last two months developing an alternative live branching assay, in collaboration with Remy Chait, a recently appointed Living Systems Institute lecturer in microfluidics. We are encouraged by our progress but, unfortunately, we are not yet at a point at which we can robustly image branched nucleation using our purified components to the quality expected for *eLife*. We also contacted Thomas Surrey, with whom we have had conversations in the past about live imaging of MTs. However, given Thomas’ imminent move to Barcelona, he was unable to assist in the timeframe required.

As an alternative, we have, therefore, re-performed the fixed analysis of dynamic branched MT nucleation, taking samples at 1, 2 and 5 minutes after initiation, fixing with glutaraldehyde and imaging. We believe these images, now incorporated into a revised Figure 3 (Figure 3B), very clearly demonstrate the progressive morphological changes of MT nucleation, showing MT branching within 2 minutes of simultaneous addition of Augmin and γ-TuRC, and leading nicely on to the latter panels, where we used pre-assembled fixed MTs and both two and three colour imaging.

2) Throughout the manuscript, the authors refer to "pure" proteins, for example "pure Augmin." However, none of the protein reagents are pure as would be revealed by analysis by mass spectrometry. Therefore "pure" should be replaced with "purified" or "enriched."

A very fair point. We have replaced these instances as asked.

Following the same reasoning, the statement "It demonstrates conclusively that Augmin directly recruits γ-TuRC to MTs and that these two protein complexes are sufficient for the phenomenon of branched MT nucleation." cannot be made. This issue could be potentially resolved by demonstrating a high efficiency of the reconstitution (see point 3 below) and by ensuring the quality of the purified complexes. The authors should at least show that they purify complexes of the expectd size (e.g. by gel filtration or sucrose gradient analysis) and also probe purified augmin with γ-TuRC antibodies and purified γ-TuRC with augmin antibodies to exclude significant co-purification. It is also important to compare size and composition of preparations from the two different cell cycle stages that are used in the assays. Gels are only shown for "interphase" augmin and γ-TuRC.

We have now more fully assessed the purity of the complexes to allow us to maintain the conclusion we drew in the original submission. As requested, we have undertaken sucrose gradient analysis for each of the two purified complexes, Augmin and γ-TuRC, for each cell cycle stage and probed purified complexes to exclude significant co-purification of Augmin and γ -TuRC. We present these in a revised Figure 1. So, in Figure 1D, we now include western blots of MG132 (mitotic) Augmin and γ-TuRC purifications, probed with reciprocal antibodies. This clearly demonstrates that subunits of neither complex co-purify with each other. In addition, in Figure 1F, we show the results of sucrose gradients undertaken from MG132 (mitotic) Augmin and γ-TuRC purifications, probed with anti-GFP. These results show that Msd1-GFP sediments as expected for Augmin-GFP (~360kD) and that γ-Tubulin-GFP sediments in two populations – one consistent with γ-Tubulin-GFP alone and one consistent with incorporation into the γ-TuRC (2MD). Neither overlap with each other, again strongly suggesting that these two complexes are purified independently – and independently of additional activities. We now mention these results in the text.

We have undertaken similar sucrose gradients for Augmin and γ-TuRC purified from interphase/cycling embryos. As they show similar profiles and as our focus here is on the activity of mitotic Augmin and γ-TuRC we have chosen not to show these in the Figure. If the reviewers feel it is important to show the similar sucrose gradient profiles, we will incorporate those panels into a further Supplementary Figure.

Finally, to clarify, the SYPRO-ruby gels shown in Figure 1 are for the “mitotic” Augmin and γ-TuRC, not interphase.

3) Please indicate protein concentrations used in each experiment in a reader-friendly manner (provide nM or µM instead of providing volume and mass).

Within the manuscript text, we have now provided the estimated protein concentrations used in each experiment in nM. We have also provided a summary of how we arrived at these concentrations within the Materials and methods, and a detailed explanation the calculations as Materials and methods subsection "Calculation of the molarity of Augmin and γ-TuRC following Cl^-^AP". The sucrose gradient results (now shown in Figure 1F; see point above) were used to determine the proportion of Msd1-GFP and γ-Tubulin-GFP incorporated into Augmin and γ-TuRC respectively. For purified Augmin, protein concentration can be directly arrived at through comparison with the concentration of GFP moiety, as all the Msd1-GFP was found to be incorporated in a complex of the expected size of Augmin. This provides an estimate of purified Augmin concentration of 300nM.

However, estimation of the concentration of γ-TuRC requires both an estimation of the proportion of γ-Tubulin-GFP incorporated into γ-TuRC – obtained from the sucrose gradients (~60% in TuRC and 40% monomeric) – and an estimation of the number of molecules of γ-Tubulin-GFP within the γ-TuRCs. Purified γ-TuRCs contain ~14 molecules of γ-Tubulin. As the γ-Tubulin-GFP expressing embryos also express endogenous γ-Tubulin, the γ-TuRCs isolated by Cl^-^AP will be heterogeneous. We have now included a western blot of post-cleaved eluate isolated from γ-Tubulin-GFP expressing embryos, probed with anti- g-Tubulin antibodies (Figure 1—figure supplement 1F). This provides an estimate that ~75% of the γ-Tubulin is present as γ-Tubulin-GFP while 25% is present as γ-Tubulin with no tag. Therefore we estimate that the purified γ-TuRCs contain, on average, 8-10 molecules of γ-Tubulin-GFP and 4-5 molecules of unlabelled γ-Tubulin. Therefore, we estimate the molar concentration of γ-TuRC purified by Cl^-^AP under the conditions described, as 300nM x 0.6/9 = 20nM.

The authors should provide an estimate of the efficiency of the branched nucleation, the fraction of γ-TuRC that is active.

We believe the co-incubation of taxol stabilised Alexa488-MTs + mitotic Augmin with Rhod-MTs + mitotic γ-TuRC, reported in our original Figure (B,C,F), provides us with the perfect experiment with which to answer this question. Based on the concentrations of the protein complexes in our assay and the number of MTs per field of view, we predict that every Rhod-MT minus end should have a γ-TuRC bound to it, and that Augmin-dependent γ-TuRC binding sites on Alexa488 MTs are in excess. Therefore, the ratio of Rhod-MTs terminating precisely at an Alexa488 MT against the total number of Rhod-MTs provides an estimate of the fraction of γ-TuRCs that are active (where activity is defined as the ability of a γ-TuRC to simultaneously bind Augmin and the minus end of a MT). We have modified Figure 3D – the histogram that originally detailed the% of Alexa488 MTs with Rhod MT branches; it now measures and represents the% of Rhod MTs that terminate at an Alexa488 MT.

This quantification demonstrates that, in the presence of Alexa488-MTs+Augmin, 55% of Rhod-MTs+γ-TuRC form branches. Moreover, when repeated using the Alexa633-GMPCPP-MT seeds, labelling the minus ends of the Rhod MTs, the percentage increases to 64%. This probably reflects a proportion of Rhod MTs that break during the preparation process, losing their native γ-TuRC-bound minus end. Thus, we conclude that the fraction of γ−TuRC that is active in our assays is at least 64%.

We have altered the manuscript to reflect the above result, commenting that while it is still possible that other proteins in the cell either further enhance the ability of g-TuRC to bind Augmin, or increase the dynamics with which it is able to nucleate MTs, it is clear that mitotic γ-TuRC, isolated from *Drosophila* embryos using Cl^-^AP is sufficient for robust and efficient branched MT nucleation in vitro.

[Editors' note: further revisions were requested prior to acceptance, as described below.]

The reviewers have evaluated your revisions and have noted that an important issue raised in the first round of review was not addressed. It concerns the estimation of the specific activity of gTuRC in the branching nucleation assay (assay in Figure 3A). As outlined in the previous decision letter and in the follow-up clarification, the reviewers feel that without this information it is difficult to assess if the reconstituted system is complete. It seems that in your revision you have provided an alternative quantification, the efficiency of branch site formation from preformed MT/augmin and MT/gTuRC complexes. However, this is not a measure of the gTuRC nucleation activity.

We agreed with the reviewers that we would count the number of MTs nucleated in the gTuRC+Augmin polymerisation assay at early timepoints, specifically t=2 and t=5 minutes. These would then directly correlate with the panels shown in Figure 3B from our previously revised manuscript. Given that, in our initial Response, we estimated the number of gTuRCs per ul of eluate, we could then relate the number of gTuRCs per field of view, with the number of MTs nucleated in that field of view. This would give a measure of nucleation activity.

Results:

Measuring the number of MTs at t=2 mins of the polymerisation assay

We obtained data from 10 fields of view from both t=2 and t=5 minutes in the polymerisation assay, counting the number of MTs using Image J. We also manually counted the number of branching MTs, providing this as a percentage of total MTs. Finally, we used Image J to automatically segment and measure the length of the MTs at each time point.

We describe this experiment in a revised section:

“Samples taken at t=2 and t=5 minutes showed similar numbers of MTs to each other, per field of view, though the mean length of MTs, and proportion of MTs with branches, were significantly greater after 5 minutes, than after 2 minutes (3.24 μm versus 4.01 μm and 34% versus 24%, respectively) (Figure 3B; Figure 3—source data 1). This suggests that MT nucleation from purified γ-TuRCs reaches maximal activity within the first few minutes of the assay”.

In the Materials and methods under “Image and Image analysis”, we detail how the above values were measured.

In the raw tabular data file, we now add the measurements within a new, appropriately labelled, sheet. We have also edited the legend for this file.

Measuring the activity of the purified gTuRCs

We then go on to use these values to calculate the activity of the gTuRCs, in the following manner:

In our previously revised manuscript, we calculated the number of molecules of GFP per ul of Cl^-^AP eluate (8.5ng) as 1.8x1011. We used this to estimate the number of molecules of purified g-TuRC, taking into account that ~9 of the g-Tubulin molecules in a purified TuRC is present as GFP-g-Tubulin (via Western blotting) and that 60% of the GFP-g-Tubulin purified is present as part of g-TuRCs (via sucrose gradient fractionation).

Therefore, we estimate that the number of molecules of g-TuRC in 1 μl of eluate is 1.8x1011 x 0.6/9 = 1.2x1010

In our polymerisation assay, we added 5 μl of GFP-g-TuRC eluate to a 50 μl final volume. At t=2 and t=5, we then took 1 μl of this reaction mix and diluted into 200 μl of glutaraldehyde fixative. We then spotted 1 μl of this onto a coverslip. Therefore, the number of g-TuRCs on the coverslip can be estimated as (1.2x1010 x 5)/1000 = 6.0x106.

Given that there was an average of 130 MTs in a field of view at t=2 mins and that the field of view was 60.5 μm2, the 22 mm2 coverslip contains ~1.32x105 fields of view; equating to 1.7x107 MTs per coverslip.

Therefore, within 2 minutes of incubation, 6.0x106 g-TuRCs are able to nucleate 61.7x107 MTs. We state this in two ways in the manuscript: (i) that one TuRC is capable of nucleating 2-3 MTs within 2 minutes, (ii) purified TuRCs nucleate MTs in the assay at ~1MT/min.

We now say: “To calculate this activity, we related the estimated number of γ-TuRCs per field of view to the mean number of MTs (see Materials and methods), These calculations suggest that a single γ-TuRC is able to nucleate multiple (2-3) MTs within 2 minutes (i.e. a rate of >1MT/min).”

In the final discussion, we state: “Although our calculations suggest that 65% of the mitotic g-TuRCs are competent to bind MT minus ends and Augmin, and that they are able to nucleate at a rate of > 1MT/min, we expect that, in vivo, other proteins either further enhance these γ-TuRC activities.”

In the Materials and methods, we add a section headed: “Estimation of the nucleation efficienct of purified g-TuRC in vitro”, providing the detailed calculations above.

We very much hope that the reviewers will look positively on this additional data.